# Exploring the cellular landscape of circular RNAs using full-length single-cell RNA sequencing

Wanying Wu[1,2,5], Jinyang Zhang [1,5], Xiaofei Cao[3], Zhengyi Cai[1] & Fangqing Zhao [1,2,3,4 ✉]

Previous studies have demonstrated the highly specific expression of circular RNAs (circRNAs) in different tissues and organisms, but the cellular architecture of circRNA has never been fully characterized. Here, we present a collection of 171 full-length single-cell RNA-seq datasets to explore the cellular landscape of circRNAs in human and mouse tissues. Through large-scale integrative analysis, we identify a total of 139,643 human and 214,747 mouse circRNAs in these scRNA-seq libraries. We validate the detected circRNAs with the integration of 11 bulk RNA-seq based resources, where 216,602 high-confidence circRNAs are uniquely detected in the single-cell cohort. We reveal the cell-type-specific expression pattern of circRNAs in brain samples, developing embryos, and breast tumors. We identify the uniquely expressed circRNAs in different cell types and validate their performance in tumor-infiltrating immune cell composition deconvolution. This study expands our knowledge of circRNA expression to the single-cell level and provides a useful resource for exploring circRNAs at this unprecedented resolution.

[1] Beijing Institutes of Life Science, Chinese Academy of Sciences, Beijing 100101, China. [2] University of Chinese Academy of Sciences, Beijing 100049, China. [3] Key Laboratory of Systems Biology, Hangzhou Institute for Advanced Study, University of Chinese Academy of Sciences, Chinese Academy of Sciences, Hangzhou, China. [4] State Key Laboratory of Integrated Management of Pest Insects and Rodents, Institute of Zoology, Chinese Academy of Sciences, Beijing, China. [5] These authors Contributed equally: Wanying Wu, Jinyang Zhang. ✉email: zhfq@biols.ac.cn

Circular RNAs are a large class of RNAs that widely exist in eukaryotic cells. Recent studies have demonstrated the emerging roles of circRNAs in regulating biological processes through promoting protein functions[1–3] or encoding peptides[4]. So far millions of circRNAs have been identified across various species, and several comprehensive databases have been developed to reveal the circRNA expression landscape in different tissues and organisms[5–7]. Generally, most circRNAs are expressed at low levels, and exhibit high tissue- and species- specificity compared to the cognate linear mRNAs[8,9]. Thus, most studies using the traditional bulk RNA-seq method cannot fully characterize the intrinsic heterogeneity between individual cells, and the complexity of circRNAs at the single-cell level needs further exploration.

The advent of single-cell RNA sequencing methods has enabled the study of the transcriptome at single-cell resolution. However, only limited attempts have been made to characterize circRNA expression patterns at single-cell resolution[10,11], which focused on studying the maternal effect of circRNAs in 69 mouse embryo samples or the heterogeneity of circRNAs among 45 single HEK293T cells. Considering the high species- and tissue-specificity of circRNAs, the cellular architecture of circRNAs in different tissues and carcinoma samples remain unexamined. Specifically, a recent study suggested that bulk RNA-seq based data were strongly affected by the cell composition in different samples, which may lead to a misleading interpretation of observed results[12]. Although most single-cell RNA-seq methods implement poly(A) selection where circRNAs should be theoretically depleted, recent studies have also demonstrated that circRNAs can be still widely detected, although with lower efficiency, in these poly(A) selected libraries, which elucidated the possibility of characterizing circRNAs using full-length scRNA-seq datasets[13–15]. Thus, the investigation of circRNAs at the single-cell level has become an emerging problem in circRNA studies.

Here, we employ a compendium of full-length single-cell RNA-sequencing datasets composed of 172,137 high-confidence cells from 171 public studies to generate a comprehensive map of circRNAs in human and mouse single cells. Through large-scale integration of these scRNA-seq datasets, we demonstrate the high cell-type specificity of circRNAs in these two species at the single-cell resolution. Particularly, we elucidate the neuron-specific expression of circRNAs in brain samples and revealed the dynamic transition between maternal and zygotic circRNA expression during embryo development. We disclose the inter- and intra-tumor heterogeneity of circRNAs in 20 breast cancer patients, where circRNAs exhibit highly similar expression in the primary and metastasis tumor from the same patient. Furthermore, we unveil the cell type-specific circRNAs expression in both species and validate the applicability of circRNAs as promising biomarkers in decomposing tumor-infiltrating immune cells using bulk RNA-seq data. We also construct the circSC online platform for exploring circRNAs expression at the single-cell level, which provides unique and useful resources for the circRNA community.

## Results

**Large-scale single-cell investigation reveals circRNAs with high cellular specificity.** To elucidate the cellular architecture of circRNAs, we collected public full-length scRNA-seq datasets from 171 studies involving 58 different human and mouse tissues or cell types (Fig. 1a and Supplementary Data 1). Considering that most 3' RNA sequencing methods were unable to detect circRNAs that lack poly(A) tails, only full-length sequencing technologies including MATQ-seq[16], Quartz-seq[17], RamDA-seq[18],

SMARTer[19], Smart-seq[20], Smart-seq2[21], SUPeR-seq[10] and Tang's method[22] were collected in our study. Then, the single-cell level expression values of both genes and circRNAs were calculated using a comprehensive pipeline embedding multiple state-of-art tools (Fig. 1b). In brief, the HISAT2[23] and StringTie[24] pipeline were used to generate the gene expression matrix, and the quality control step was utilized to filter high-confidence cells using the Scater[25] package. To eliminate the batch effect across different studies, the anchor-based canonical correlation analysis (CCA) method in Seurat[26] package was performed, and cells were clustered using principal component analysis and k-nearest neighborhood clustering. Then, cell clusters were annotated using published results and manually curated using cell markers. At the same time, the single-cell expression matrix of circRNAs was obtained using the CIRI2[27,28] and CIRIquant[29] pipeline, and circRNAs with at least 2 supporting reads were kept for the downstream analysis. Then, the expression level of circRNAs was consequently normalized using gene expression profiles (see Methods). In summary, 40,604 human and 131,533 mouse single cells passed quality control (approximate 1000 cells per experiments), and circRNAs in these cells were detected for downstream analysis.

To evaluate the reliability of circRNA detection, all circRNAs in single-cell data were comprehensively compared against our previous database circAtlas v2.0[7] or the integration of other 10 bulk RNA-seq based circRNA databases (Supplementary Table 1). Considering that only the circAtlas database provides the assembled full-length sequence and conservation score of reported circRNAs[30,31], the circRNA set obtained from the circAtlas database was analyzed separately. As shown in Fig. 1c, a total of 354,390 circRNAs were detected in the scRNA-seq cohort, where 76,824 (21.67%) circRNAs can be simultaneously detected in all three circRNA sets (Supplementary Fig. 1a, b). In summary, 32.43% of circRNAs were also present in these bulk RNA-seq databases, while the remaining 67.57% of the circRNAs were uniquely detected in single-cell data. Notably, circRNAs that were uniquely detected in circAtlas have significantly lower expression levels (measured by counts per million, CPM) and shorter lengths than those shared in both circAtlas and single-cell datasets (Fig. 1d, e), indicating that scRNA-seq can effectively capture most high-abundance circRNAs. Besides, these shared circRNAs exhibited high tissue specificity measured by MCS score according to our previously described method[7]; 48.9% of these overlap circRNAs were conserved across more than two species (MCS score ≥2), demonstrating the high reliability of our identified circRNAs (Fig. 1f).

For all circRNAs detected in the scRNA-seq datasets, a positive correlation ($R = 0.53$) between the number of expressing cells and their mean expression level were detected (Fig. 1g and Supplementary Fig. 1c), and several highly-expressed circRNAs like mmu-Cdr1_0001, mmu-Tulp4_0006, and hsa-RIMS1_0021 were also reported in previous studies[32–34], which further supported the circRNA identification results. Meanwhile, circRNAs that were uniquely detected in scRNA-seq data were generally expressed in a lower number of cells (Fig. 1h, $p < 0.001$, Wilcoxon rank-sum test) but have similar expression levels compared to circRNAs validated by other databases (Fig. 1i, $p = 0.09$, Wilcoxon rank-sum test), suggesting the high cell-specific expression of these circRNAs. Specifically, ~90% of scRNA-seq specific circRNAs were expressed in less than 10 cells in both human and mouse samples, which makes it almost impossible to be detected using bulk RNA-seq techniques (Fig. 1j and Supplementary Fig. 1d). Taken together, these results indicated the high sensitivity and reliability of full-length scRNA-seq to reveal circRNAs with high cell specificity, while most of which could be falsely neglected due to the relatively

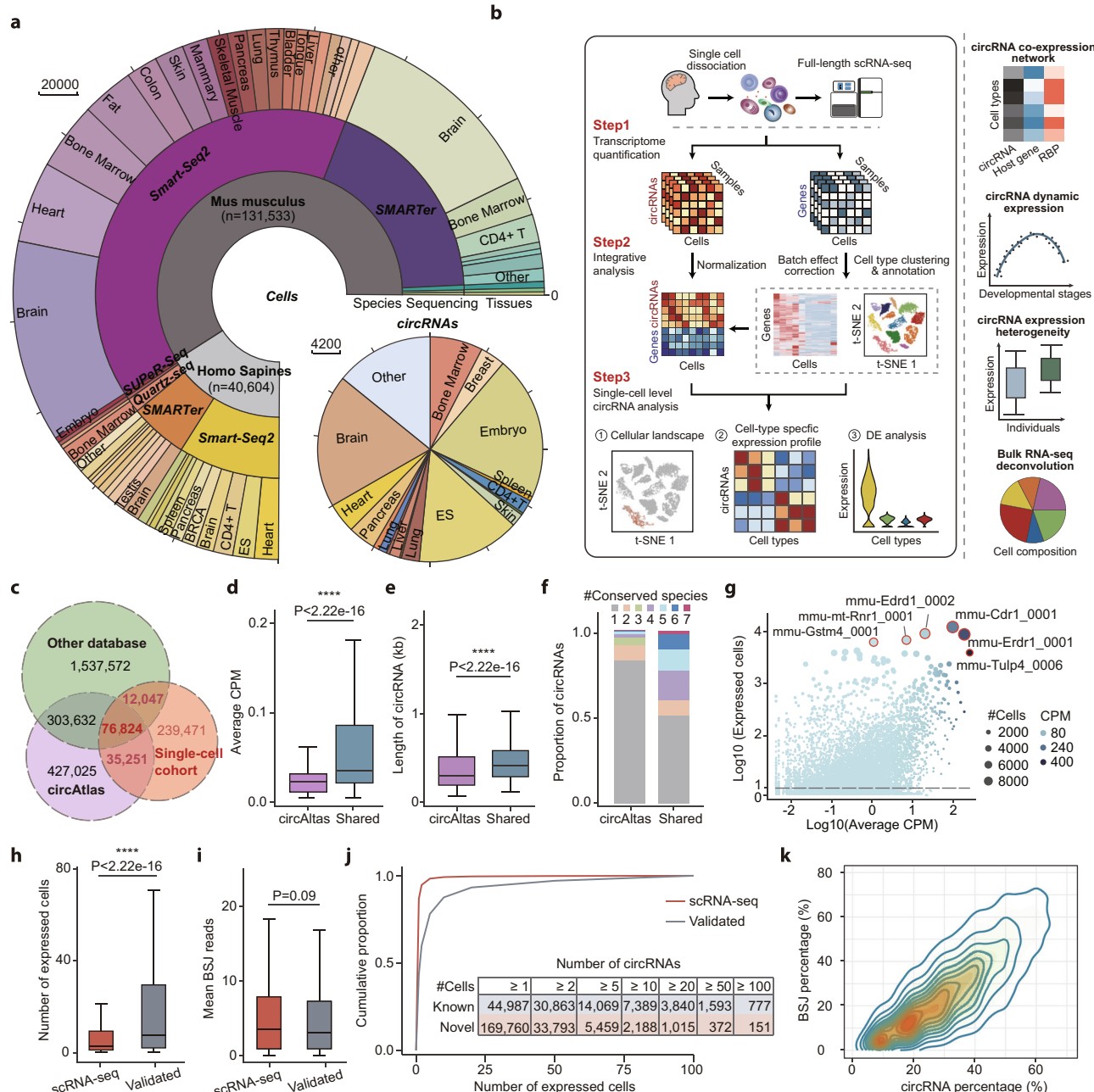

**Fig. 1 Discovering circRNAs from large-scale full-length single-cell RNA-seq datasets. a** The total number of cells and circRNAs detected in the collected scRNA-seq datasets. **b** Workflow of scRNA-seq data integration and circRNA detection (see Methods). **c** Overlap of circRNAs detected in the scRNA-seq datasets, circAtlas database, and the integration of other 10 bulk RNA-seq based databases. **d** The average expression levels (counts per million, CPM) of circRNAs in the circAtlas database. Colors represent circRNAs that were uniquely detected in circAtlas (purple, $n = 730,657$) or simultaneously detected in circAtlas and the scRNA-seq cohort (blue, $n = 112,075$). **e** The length of fully assembled sequence of 619,060 circAtlas-specific and 103,758 circRNAs shared between circAtlas and the scRNA-seq cohort. **f** The number of species that circRNAs were conservatively expressed in. **g** Log-scaled mean expression level and the number of expressing cells for mouse circRNAs. Sizes of points indicate the number of expressing cells. Filled colors represent the mean CPM of circRNAs. **h, i** The number of circRNA host cells (**h**) and mean BSJ reads (**i**) of circRNAs ($n = 239,471$) that were uniquely detected in the scRNA-seq data or validated circRNAs ($n = 114,919$) which were also observed in bulk RNA-seq circRNA databases. **j** Cumulative distribution of mouse circRNAs ordered by the number of expressing cells. The scRNA-seq specific and validated circRNAs are colored in red and grey, respectively. **k** Expression of scRNA-seq specific circRNAs in cells that have more than 10 BSJ reads in total. The *x*- and *y*-axis represent the proportion of scRNA-seq specificcircRNA number and BSJ reads in these cells, respectively. Colors indicate the density of circRNAs at each point. All center lines in the box plots indicate the median values, and box limits indicate the upper and lower quartiles of plotted values. The upper and lower whiskers indicate the largest and smallest values within the range of 1.5x interquantile range (IQR) distance from the box limits. ****$P < 0.0001$, Wilcoxon rank-sum test (two-sided). Source data are provided as a Source Data file.

lower proportion of expressing cells in traditional bulk RNA-seq samples. Additionally, these scRNA-seq specific circRNAs were also widely expressed in cells that have more than 10 back-spliced junction (BSJ) reads (Fig. 1k). Despite the small number of expressing cells, these circRNAs were also originated from exons that have higher conservation scores (Supplementary Fig. 1e, f). Besides, a proportion of 16.0% human and 5.0% mouse scRNA-seq specific circRNAs also exhibited conservative expression in more than two species (Supplementary Fig. 1g), which suggested that a large fraction of conserved circRNAs with potential biological functions remain undiscovered in the previous bulk RNA-seq datasets.

**Brain circRNAs display cell-specific expression patterns in inhibitory and excitatory neurons**. Previous studies have shown that circRNAs are widely expressed across eukaryotic tissues, and especially enriched in mammalian brains[8,9,35,36]. However, the cellular expression pattern of circRNAs in this tissue has never been examined. To rigorously investigate the cellular landscape of circRNAs, we first collected and analyzed 18 studies of mouse brain samples, which also constitute the largest cohort among our collected datasets. All human and brain cells were analyzed and integrated as described above. A total of 41,911 cells were divided into 14 clusters, and 64,311 circRNAs were detected (Fig. 2a). As shown in Fig. 2b, most cells were clustered into GABAergic neurons (GABA), glutamatergic neurons (GLUT), and microglia cells (MG). Despite the similar number of cells in these clusters, circRNAs tend to be specifically enriched in GABAergic and glutamatergic neurons. Although the top 10 most abundant circRNAs have shown conserved expression across different cell types, cell-type-specific circRNAs exhibited disparate patterns between neurons, immune cells, glial cells, and vascular cells, demonstrating the high cell specificity of these circRNAs. For experimental validation of these circRNAs, RT-PCR of 12 cell-type-specific circRNAs that expressed in less than 10 cells were performed using outward primers targeting the BSJ region, and the back-spliced junction sequence of these circRNAs were successfully validated using Sanger sequencing (Supplementary Table 2). Then, the widely used *Tau* method[37] was implemented to measure the cellular specificity of circRNAs, and genes were divided into circRNA hosting genes and other genes for further comparison. As shown in Fig. 2c, circRNAs exhibited a significantly higher specificity than both groups of genes. Meanwhile, the circRNA hosting genes also showed significantly lower specificity than other non-hosting genes, as that circRNAs tend to be originated from genes with higher expression levels (Supplementary Fig. 2a), which resulted in a relatively lower cell specificity. For instance, 10 of 12 circRNAs from the mouse *Taf1* gene were specifically detected in neuron cells, and a distinct expression pattern was also observed in GABAergic and glutamatergic neurons (Fig. 2d).

To further validate the circRNAs expression landscape in the human brain, four scRNA-seq datasets (GSE67835, GSE71315, GSE75140, and GSE125288) of healthy human brains were also analyzed, and the enriched expression of circRNAs in GABAergic and glutamatergic neurons was observed accordingly (Supplementary Fig. 2b, c). Afterward, the orthologs between human and mouse circRNAs were extracted from the circAtlas database. As shown in Fig. 2e, circRNAs with higher expression levels were more likely to be conserved in both species, whereas species-specific circRNAs tend to have lower expression levels. Consistent with previous results, the majority of these conserved circRNAs were highly enriched in GABAergic and glutamatergic neurons, and a proportion of circRNAs were also exhibited to be generally expressed in all types of cells (Fig. 2f). As mentioned in previous

studies, the expression level of circRNAs is largely correlated to the activity of RNA-binding proteins (RBP)[9,38,39]. Thus, to explain these patterns, the Spearman correlation coefficient between all circRNAs and circRNA hosting genes or RBPs in all cells was calculated for comparison. The correlation coefficient between circRNAs and RBPs was significantly higher ($p < 0.001$) than that of hosting genes (Fig. 2g). In particular, the polypyrimidine tract binding proteins PTBP1 ($R = 0.76$) and PTBP2 ($R = 0.66$) exhibited a high correlation against circRNAs, where a relatively low level of PTBP1 and a high level of PTBP2 were observed in both GABAergic and glutamatergic neurons. Our previous study has shown that the decrease of PTBP1 activity could result in a dramatic outburst of circRNAs[29], which can partially explain the enormous number of neuron-specific circRNAs detected in the single-cell cohort. As expected, the circRNA expression level (e.g., circCdr1) and circular-to-linear ratio were highly correlated with the downregulation of PTBP1 and the upregulation of its compensator PTBP2 in most cell types (Fig. 2h). Furthermore, only a small proportion of overlap between circRNA-generating loci in GABAergic and glutamatergic neurons was observed (Supplementary Fig. 2d), which indicated the cell-specific expression in these two types of neurons. The gene ontology analysis also demonstrated the enrichment of excitatory synapse and glutamate decarboxylase complex in GABAergic- and glutamatergic-specific circRNAs, which is consistent with the biological characteristic of GABAergic and glutamatergic neurons, respectively (Supplementary Fig. 2h). Taken together, these results demonstrate the highly cellular-specific expression landscape of circRNAs, and further reveal the complex association between circRNA biogenesis and RBP activity, especially in these inhibitory and excitatory neurons.

**The dynamic expression of maternal and zygotic circRNAs during early embryo development**. Single-cell RNA sequencing has enabled the study of gene heterogeneity in embryonic development stages[40], but the change of circRNA expression pattern during this process still needs further exploration. Here, we analyzed 11 studies of human and mouse embryos containing samples from 16 different stages covering oocytes to early buds (Fig. 3a). A total of 41,041 and 24,818 circRNAs were detected in human and mouse embryonic cells, respectively. To reveal the dynamic changes between circRNAs in the embryo developing process, the Pearson correlation between circRNA expression levels in different stages was calculated. As shown in Fig. 3b, a high correlation between cells in the first 3-4 days after fertilization was observed, which is consistent with the maternal effect of circRNAs during early embryonic development[10,41]. Moreover, cells from blastocyst to implanted embryos exhibited a different expression pattern of circRNAs, suggesting the expression of zygotic circRNAs after the blastocyst stage. Besides, an increase in both the circRNA diversity and junction ratio of detected circRNAs within developing stages were observed on both human and mouse samples, which also verified the accumulation of these zygotic circRNAs in the embryo developing process (Fig. 3c and Supplementary Fig. 3a). Considering that only a relatively small number of cells were collected in the human datasets, only mouse embryos were included in the downstream analysis. To eliminate the randomness effect, the expression pattern of circRNAs that can be detected in more than two stages was plotted in Fig. 3d. As expected, the gradual degradation of maternal circRNAs was observed, and most other circRNAs exhibited a stage-specific expression profile. To further investigate the dynamic expression changes of circRNAs during the maternal-to-zygotic transition, samples were divided into four-time points including totipotent

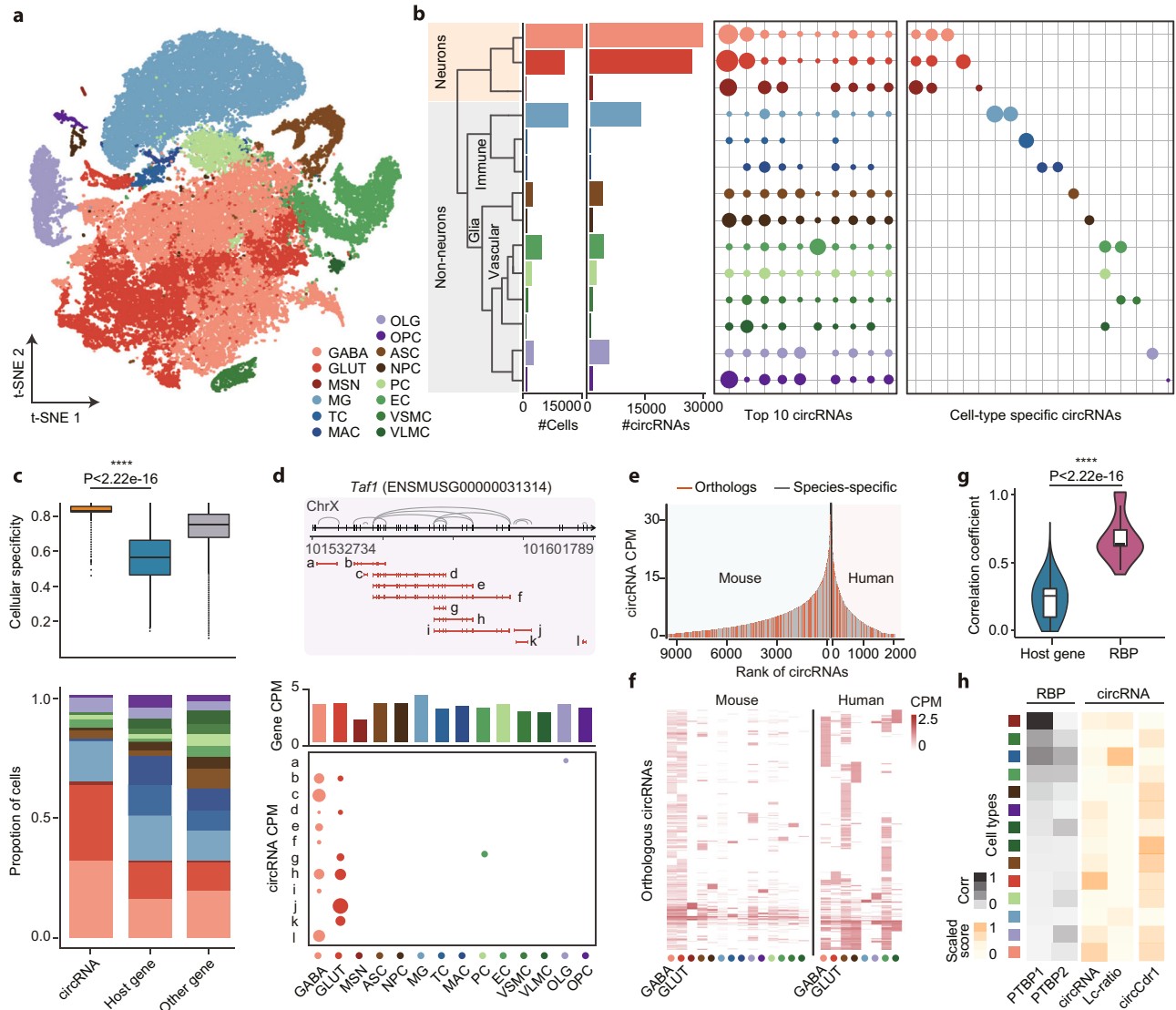

**Fig. 2 Enriched expression of circRNAs in inhibitory and excitatory neurons. a** The t-SNE plot of all 41,911 brain single cells, colored by annotated cell types. GABA, GABAergic neurons. GLUT, glutamatergic neurons. MG, microglia cells. **b** Panels from left to right: the number of cells; the number of identified circRNAs; expression values of top 10 highly expressed circRNAs; expressed values of cell-type-specific circRNAs in each cluster. Sizes of points indicate the mean expression values measured by CPM. **c** The cellular specificity of circRNAs ($n = 74,678$), circRNA hosting genes ($n = 13,467$), and non-hosting genes ($n = 39,253$). Upper, the cell specificity was measured by the *Tau* method. Bottom, the proportion of cell types in all expressing cells. **d** Example of 12 circRNAs generated from the *Taf1* loci. Upper, schematic view of gene structure and back-splicing events (grey lines). Bottom, mean expression levels of circRNA isoforms and *Taf1* gene in each cell cluster. **e** Orthologues between human and mouse circRNAs. All human and circRNAs were ranked according to their expression levels, and the *y* axis represents the average CPM of circRNAs in all expressing cells. Red, circRNAs that are conservatively expressed in human and mouse. Grey, circRNAs uniquely detected in each species. **f** Expression heatmap of 1,048 orthologous circRNAs in human and mouse cells. Filled colors represent the mean CPM of circRNAs in each cluster. **g** The Spearman correlation between circRNAs and host genes ($n = 13,467$) or RBPs ($n = 2995$). **h** Cell-type-specific expression of RBPs and circRNAs. Filled colors indicate the normalized value of gene or circRNA expression values in each cell type. All center lines in the box plots and violin plots indicate the median values, and box limits indicate the upper and lower quartiles of plotted values. The upper and lower whiskers indicate the largest and smallest values within the range of 1.5x IQR from the box limits. ***$P < 0.001$, Wilcoxon rank-sum test (two-sided). Source data are provided as a Source Data file.

blastomeres (TB), first lineage (TE/ICM), second lineage (EPI/PE), and implanted embryo, reflecting the changes of totipotency and lineage segregation in the development process. Subsequently, genes and circRNAs were clustered into 5 groups using a noise-robust clustering method[42]. As shown in Fig. 3e, circRNAs and genes in cluster 1 and 2 were highly expressed in the early TB stage, then continuously decreased with the embryo development. In contrast, cluster 3 to 5 of circRNAs represent zygotic circRNAs that were specifically expressed after fertilization. To determine whether the activation of zygotic circRNAs were byproducts of

host gene expression, the correspondence between circRNAs and their host genes was examined. Notably, a large fraction of zygotic circRNAs (67.50% in cluster 3, 69.2% in cluster 4, and 83.9% in cluster 5) were generated from maternally expressed genes, which suggested the unique biogenesis mechanism of these zygotic circRNAs during embryo development (Fig. 3h).

To further investigate the difference between the zygotic gene and circRNA activation process, the composition of reads from genes and circRNAs in each cluster was calculated. Similarly, only circRNAs that simultaneously expressed in more than one of four

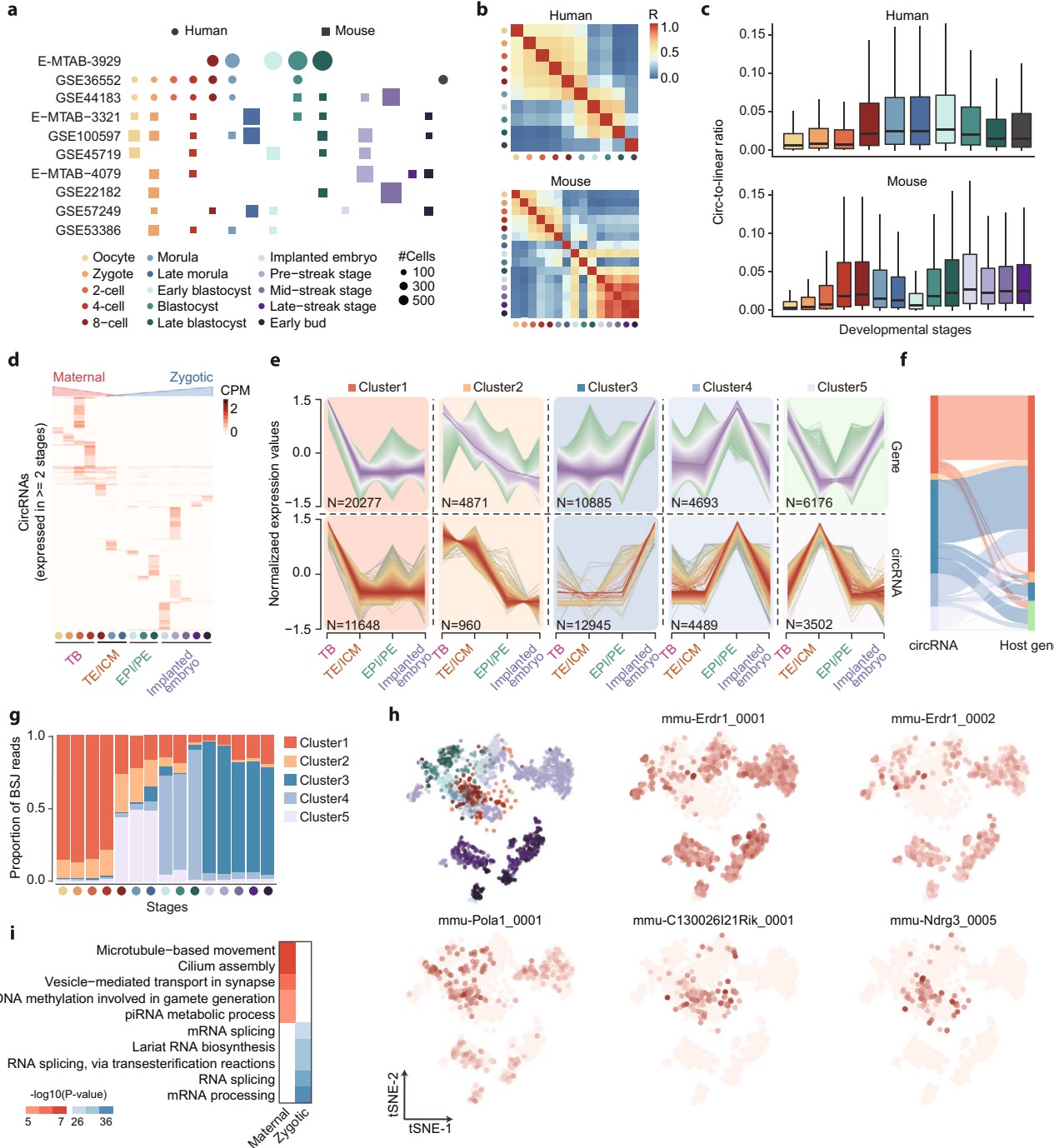

**Fig. 3 Dissection of zygotic circRNA activation during the maternal-to-zygotic transition. a** Summary of scRNA-seq datasets on different embryo developing stages of human and mouse. The size of points indicates the number of cells in each developing stage. **b** Correlation of circRNAs expression in human and mouse embryo developing stages. Filled colors represent the Pearson correlation coefficient between average expression levels of circRNAs in each stage. **c** Boxplot illustrating the circular-to-linear ratio of circRNAs in each stage of human (*n* = 5876/ 11,908 / 13,561 / 18,634 / 16,379 / 22,717 / 4,090 / 9,160 / 2,573 / 2,784 circRNAs) and mouse (*n* = 1928 / 2579 / 10,998 / 6789 / 3174 / 3262 / 62 / 1142 / 1306 / 2279 / 2997 / 7300 / 805 / 1079 / 1300 circRNAs) embryos. The y axis represents the circular-to-linear ratio measured as the expression level of circRNAs divided by the host gene expression values. The center lines indicate the median values, and box limits indicate the upper and lower quartiles of plotted values. The upper and lower whiskers indicate the largest and smallest values within the range of 1.5-fold IQR from the box limits. **d** Heatmap of circRNAs (expressed in ≥ 2 stages) expression. Filled colors represent the average CPM of circRNAs in each stage, and all stages were divided into four time points according to the cell differentiation and lineage segregation state. **e** Fuzzy clustering of circRNAs expression data in four time points. Purple or red colored lines correspond to genes with high membership value, and y axis represents the normalized expression value from the Mfuzz result. **f** The consistency of cluster results of circRNAs and host genes. Filled colors indicate the clustered results. **g** Composition of maternal (cluster 1–2) and zygotic (cluster 3–5) circRNAs in each stage. **h** The t-SNE projection of all cells and typical maternal (bottom) and zygotic (upper) circRNAs, colored by indicated developing stages points or expression level of circRNAs. **i** Gene ontology enrichment for the maternal and zygotic circRNAs. Colors indicate q-value computed using two-sided Fisher exact test and adjusted using Benjamini-Hochberg method for multiple hypotheses testing. Source data are provided as a Source Data file.

stages were included. In contrast to the gentle increase of zygotic gene reads during the developing stages (Supplementary Fig. 3b), the dramatic outbreak of zygotic circRNAs after 8-cell stages was observed in Fig. 3g, providing convincing evidence of maternal circRNA degradation and zygotic circRNA activation. For instance, the different expression patterns of two zygotic and three maternal circRNAs were plotted. As shown in Fig. 3h, the mmu-Erdr1_0001 and mmu-Erdr1_0002 derived from erythroid differentiation regulator-1 (Erdr1), a secreted factor that regulates cell survival, apoptosis[43,44], were highly expressed in the implanted embryo. Meanwhile, the Erdr1 gene was lowly expressed in cells from all stages, suggesting that the possible biological function of mmu-Erdr1_0001 and mmu-Erdr1_0002 in the development of the implanted embryo (Supplementary Fig. 3c). Moreover, mmu-Pola1_0001, mmu-C130026I21Rik_0001, and mmu-Ndrg3_0005 also exhibited a stronger maternal effect compared to their host genes. Thus, the highly specific expression of these circRNAs demonstrated that circRNAs undergo a more significant maternal-to-zygotic transition process compared to their linear counterparts. Finally, the gene ontology enrichment analysis was performed on the parental gene of maternal and zygotic circRNAs. As shown in Fig. 3i, microtubule-based movement and cilium assembly were enriched in the maternal circRNAs, while splicing-related processes were enriched in the zygotic circRNAs, which is consistent with the polarity establishment and embryonic genome activation in developing embryos. Collectively, these results demonstrated the highly cellular specific expression profile of circRNAs and the substantial activation of zygotic circRNAs in embryo development, which also suggested the important role of these maternal and zygotic circRNAs during this process.

**Inter- and intra-tumor circRNA heterogeneity in human breast cancer metastasis.** Recent studies have demonstrated the emerging role of circRNAs in regulating cancer progression and proliferation[45–48]. However, the comprehensive landscape of circRNA expression at the single-cell level has not been thoroughly examined. To extensively profile circRNAs across breast cancer tumorigenesis, a total of 26 primary and metastasis tumor scRNA-seq samples from 20 breast invasive carcinoma (BRCA) patients with different luminal stages including 19 TNBC, 3 HER2 negative, 2 luminal A, and 2 luminal B samples were investigated[49–51]. Firstly, all cells from 20 patients were integrated and analyzed as described above, and CopyKAT[52] was performed to identify normal cells and tumor cells with copy number variations (Fig. 4a). As shown in Fig. 4b, more than 49.88% and 67.28% of normal and carcinoma populations were identified as epithelial cells. Then, the difference of circRNA expression levels between normal and carcinoma populations was further investigated (Fig. 4b and Supplementary Fig. 4a). Consistent with previous studies, tumor cells with aneuploid rearrangement exhibited significantly lower expression of circRNAs in both metastasis and primary tumors (Fig. 4c), and the same pattern was also observed in most identified cell types (Fig. 4d). In particular, the expression of several well-known circRNAs was plotted in Supplementary Fig. 4b, whereas cancer-related circRNAs like hsa-CDYL_0005, has-BARD1_0006, hsa-HIPK3_0001, and hsa-FAM120A_0006 can be successfully detected in both normal and carcinoma cells[53–56]. Besides, a cell-specific expression pattern of circRNA isoforms derived from BARD1 and KRD36C gene were also plotted, indicating the sparse expression of circRNAs in scRNA-seq data (Supplementary Fig. 4c). Interestingly, both normal and carcinoma cells from low-grade (luminal A, luminal B, and HER2-negative) tumors with better prognosis tended to express more circRNA than high-grade triple-negative breast carcinoma

(TNBC) cells, indicating the less accumulation of circRNAs in TNBC cells with faster progression rate.

Given the dominant number of epithelial cells in this cohort and the important role of epithelial to mesenchymal transition (EMT) in tumor invasion and metastasis, the circRNA dynamic during EMT was further investigated. Firstly, all epithelial cells were clustered, and trajectory inference analysis was performed to reveal the dynamic cell differentiation process (Fig. 4f). To better explore the transition state of individual cells, the EMT score was consequently calculated using a reported method[57]. As shown in Fig. 4g, the cell trajectory results generally fitted the increase of EMT score accordingly. Then, gene ontology (GO) enrichment analysis was performed on each cell cluster. As expected, epithelial cells proliferation processes were enriched in clusters with lower EMT scores, while cell migration and mesenchymal related processes were enriched in the clusters with higher EMT levels. Furthermore, the proportion of carcinoma cells in each cluster was calculated, and a positive correlation between tumor cell percentage and EMT score was observed accordingly (Fig. 4h). This result can be explained as the EMT score was calculated using a cancer specific EMT signature matrix. Interestingly, after the intermediate EMT state (branch point 1 in Fig. 4f), epithelial cells were differentiated into two branches. The upper branch, which mainly consisted of cluster 10-12, had a significantly more proportion of carcinoma cells and a higher EMT score compared to the other branch that was made up of more normal cells. Finally, the circRNA expression level in each cluster was calculated (Supplementary Fig. 4d). With the transition from the epithelial cell (cluster 1-2) to the intermediate EMT state (cluster 3-5), the average expression level of circRNA increased accordingly (Fig. 4i), which is consistent with the global activation of circRNAs during EMT[38]. However, in the later stage of EMT, an unexpected decrease in circRNA expression was observed. In particular, the circRNA expression level in carcinoma cells was decreased in the mesenchymal stage (cluster 9-12) compared to that of normal cells. The difference between normal and carcinoma cells in the mesenchymal stage suggested the weakened accumulation effect of circRNAs with tumor cell proliferation in the later stage of EMT. Finally, the heterogeneity of circRNA between patients was investigated. As shown in Supplementary Fig. 4e, the metastasis and primary tumor from one patient exhibited similar expression patterns, and a large variation in cells from different patients could be observed. Taken together, we profiled the detailed profile of circRNA expression during EMT, revealing the complex inter- and intra-tumor heterogeneity of circRNAs between primary and metastasis samples from breast cancer patients.

**Cell-specific circRNAs providing insights into optimal cell type discrimination.** In previous studies, many computational methods have been developed to explore the heterogeneity of tumor-infiltrating immune cells in bulk RNA-seq datasets using cell type-specific marker genes[58–61]. Based on the high cellular specificity of circRNAs, we further speculated the possibility of using circRNAs as biomarkers to improve the performance of cell type decomposition. To construct a high-quality circRNA signature matrix, the scRNA-seq cohort from 17 different human and mouse tissues along with cognate cancer samples were investigated (Fig. 5a). Then, we also collected 446 and 777 bulk normal and tumor RNA-seq datasets from the circAtlas[7] and MiOncoCirc[62] databases to validate the performance of circRNA in cell-type deconvolution. In brief, all scRNA-seq samples were analyzed using the Seurat[26] pipeline (see Methods). Next, cell composition in human and mouse datasets were predicted using marker genes from published databases[63,64] and literature, and

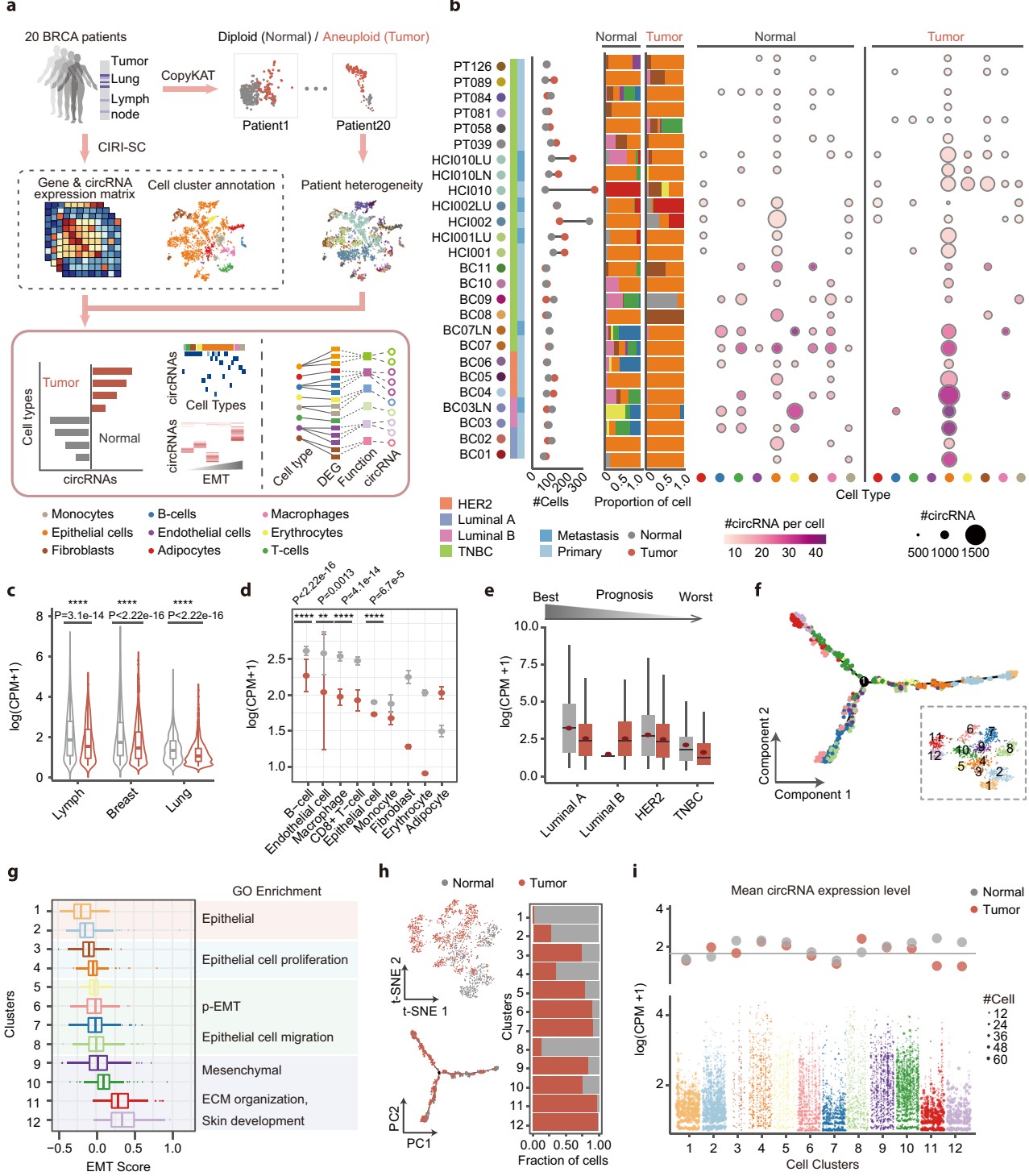

then curated using SingleR[65] prediction results. Firstly, all circRNAs were divided into five groups according to the expression pattern in different cell types and tissues (Fig. 5b). In summary, a total of 12,625 circRNAs across all samples were only detected in one cell type, of which 6,623 (52.5%) were also reported in the bulk RNA-seq based resource (Supplementary Fig. 5a). As shown in Supplementary Fig. 5b, these circRNAs were mutually detected in a variety of tissues and samples, indicating the potential of these circRNAs as biomarkers for cell-type classification. Besides, 3.24% of circRNAs were detected in multiple cell types within a single tissue, which also validated the tissue-specific expression pattern of these circRNAs. About 50% of circRNAs exhibited a

constitutive expression in more than 50% of cells or expressed in multiple cell types, suggesting the "housekeeping" role of these circRNAs in specific tissues or cell types. Meanwhile, the majority of circRNAs were "lowly expressed" in only one cell, which is consistent with the randomized biogenesis of most circRNAs reported in a recent study[66]. Afterward, the cell-type-specific architecture of circRNAs in human and mouse samples was summarized, and the relationship of shared circRNAs was plotted in Fig. 5c. Similar to the gene expression landscape reported in the previous study[67], circRNAs also exhibited distinct expression clusters between cell types with different functions. Specifically, several orthologous cell-type-specific circRNAs between human

**Fig. 4 Heterogeneity of circRNAs between normal and tumor cells in breast cancer patients. a** Schema of data integration and tumor cell identification. All cells were divided into normal and tumor cells using CopyKAT, then cells were integrated and clustered based on the expression profiles. **b** The cell number (left column), cell composition (middle column), and the mean number of circRNAs in each cell (right column). Samples were divided into normal and tumor cells according to the copy number variation. **c** Log-scaled circRNA expression values in normal and tumor cells from the primary (breast, $n = 4687/8246$ cells) and metastasis tumors (lymph, $n = 1679/1465$ cells and lung $n = 614/1770$ cells). Grey and red lines indicate normal and tumor cells, respectively. **d** Log-scaled circRNA expression values in each cell type, grey and red color indicates normal ($n = 660 / 28 / 660 / 885 / 3339 / 140 / 205 / 970 / 93$) and tumor ($n = 29 / 2 / 108 / 36 / 9,630 / 185 / 792 / 533 / 167$) cells. The error bars indicate ± SD of plotted values. **e** Log-scaled circRNA expression values divided by molecular subtypes. The $x$ axis indicates molecular subtypes ranked from best to worst prognosis. Filled colors indicate normal ($n = 518 / 1393 / 536 / 4533$) and tumor ($n = 852 / 1081 / 2926 / 6622$) cells, respectively. The red points indicate the mean value of plotted data. **f** Trajectory reconstruction of all epithelial cells reveals two branches in tumor progression, colored by cluster results from the t-SNE plot. **g** GO enrichment analysis of 12 cell clusters ($n = 149 / 239 / 62 / 143 / 89 / 165 / 209 / 200 / 119 / 183 / 159 / 126$ cells) ordered by the EMT score. All clusters were divided into four stages according to the enriched biological processes. **h** The distribution of tumor and normal cells in t-SNE and trajectory projection plots (left), and cell composition in each cluster (right). **i** Change of circRNAs expression profiles in EMT cluster. The $y$ axis represents log-scaled expression values of circRNAs, and size of points indicates the number of expressing cells. All center lines in the box plots and violin plots indicate the median values, and box limits indicate the upper and lower quartiles of plotted values. The upper and lower whiskers indicate the largest and smallest values within the range of 1.5x IQR from the box limits. **$P < 0.01$, ****$P < 0.0001$, Wilcoxon rank-sum test (two-sided). Source data are provided as a Source Data file.

and mouse cells were also detected, implying the conserved biological function of these circRNA subsets.

To validate the potential of circRNA serving as cell type biomarkers, the overlap between expressed circRNAs in different cell types and bulk RNA-seq datasets were further calculated. As shown in Fig. 5d, circRNA detected in bulk RNA-seq data exhibited a highly specific overlap with cellular expressed circRNAs. For instance, 39.36% of circRNAs detected in GABAergic neurons can be simultaneously detected in normal brain samples, and the overlap of circRNAs in human and brain samples was also highly enriched in cell types identified in the previous results. To compare the performance of circRNAs and genes as cellular biomarkers in profiling tumor-infiltrating cells, only cell types that were annotated in human tumor samples were included in the downstream analysis. Afterward, the cell-type specificity of all expressed circRNAs, marker genes from public databases, and 1,000 randomly selected genes were calculated. Notably, the cell type specificity of circRNAs was significantly higher than that of marker genes and random control genes, which further indicated the ability of circRNAs to serve as cell-type biomarkers (Fig. 5e). Then, the composition of tumor-infiltrating immune cells in cancer-related bulk RNA-seq datasets was calculated using CIBERSORT[68] with marker genes from the LM22 gene set and cell-type-specific circRNAs from immune cells, respectively (Fig. 5f). The performance of cell-type decomposition was assessed by log-scale root-mean-square error (RMSE) provided in the CIBERSORT results, which represent the bias between original and imputed marker gene expression values. As shown in Fig. 5g, the deconvolution results using circRNAs have significantly lower RMSE values ($p = 0.015$, Wilcoxon test), which represents better accuracy in estimating cell compositions. These results demonstrated the applicability of circRNAs serves as better cell-type biomarkers in exploring the heterogeneity of tumor-infiltrating immune cells, which also suggested the important biological roles of these circRNAs in certain cell types.

To this end, we further integrated the cellular architecture of circRNAs and the circRNA signature matrix in immune cells into a web server called the circRNA single-cell portal (circSC). The circSC portal provides comprehensive information including cellular expression profile, differentially expressed results, and the catalogue of circRNAs identified in an enormous number of human and mouse cells (Fig. 6). The circSC portal has been integrated into circAtlas as an individual module (http://circatlas.biols.ac.cn/), providing convenient browsing and searching functions of both the single-cell and bulk RNA-seq expression

pattern of circRNAs of interest. Thus, we believe that our database can serve as an important resource for exploring the dynamic changes of circRNAs in embryo development, tissue differentiation, and cancer biogenesis process, and it provides a unique and useful platform for the circRNA community.

## Discussion

In this study, we reported the single-cell landscape of circRNAs using a large-scale full-length scRNA-seq cohort. We identified a total of 139,643 and 214,747 circRNAs in human and mouse single cells, respectively. We also validated detected circRNAs using public resources based on bulk RNA-seq data and discovered 216,602 high-confidence circRNAs ($\geq 5$ supporting reads) that were uniquely detected in the single-cell cohort. Based on these datasets, we rigorously investigated the single-cell expression pattern of circRNAs in different tissues, developing stages, and cell states. Furthermore, we revealed the relatively higher cell specificity of circRNAs compared to the linear mRNAs and demonstrated the promising role of circRNAs in improving the performance of cell composition estimation from bulk RNA-seq datasets.

Given that circRNAs do not have poly(A) structures like their cognate linear RNAs, the most widely used oligo(dT) priming methods could not detect circRNAs effectively. However, recent studies have demonstrated the circRNAs could also be detected at low levels in poly(A) enriched libraries[14], which further validated the feasibility of studying the single-cell circRNA landscape using the tremendous number of public scRNA-seq datasets. As most circRNAs are derived from exonic regions, the identification of circRNAs largely relies on the detection of back-splicing junction sequences. Thus, most 3' end sequencing methods like Drop-seq and 10X Genomics Chromium Single Cell 3' method are not likely to generate fragments spanning the junction site and thus are not suitable for circRNA detection. Therefore, only datasets generated from 8 full-length scRNA-seq methods were collected in our study, which provided the basis for exploring circRNA expression with an unprecedented resolution.

The large-scale integration of scRNA-seq datasets provides an opportunity to reveal the dynamic changes of circRNAs in different cell types or developing stages. In this study, we found that circRNAs were highly enriched in neurons compared to other cells in brain samples. The inhibitory and excitatory neurons also exhibited cell-specific circRNA expression patterns that were correlated with RBP expression levels, suggesting the highly specific expression of circRNAs under the regulation of RBP in diverse cell types. We also explored the dynamic changes of

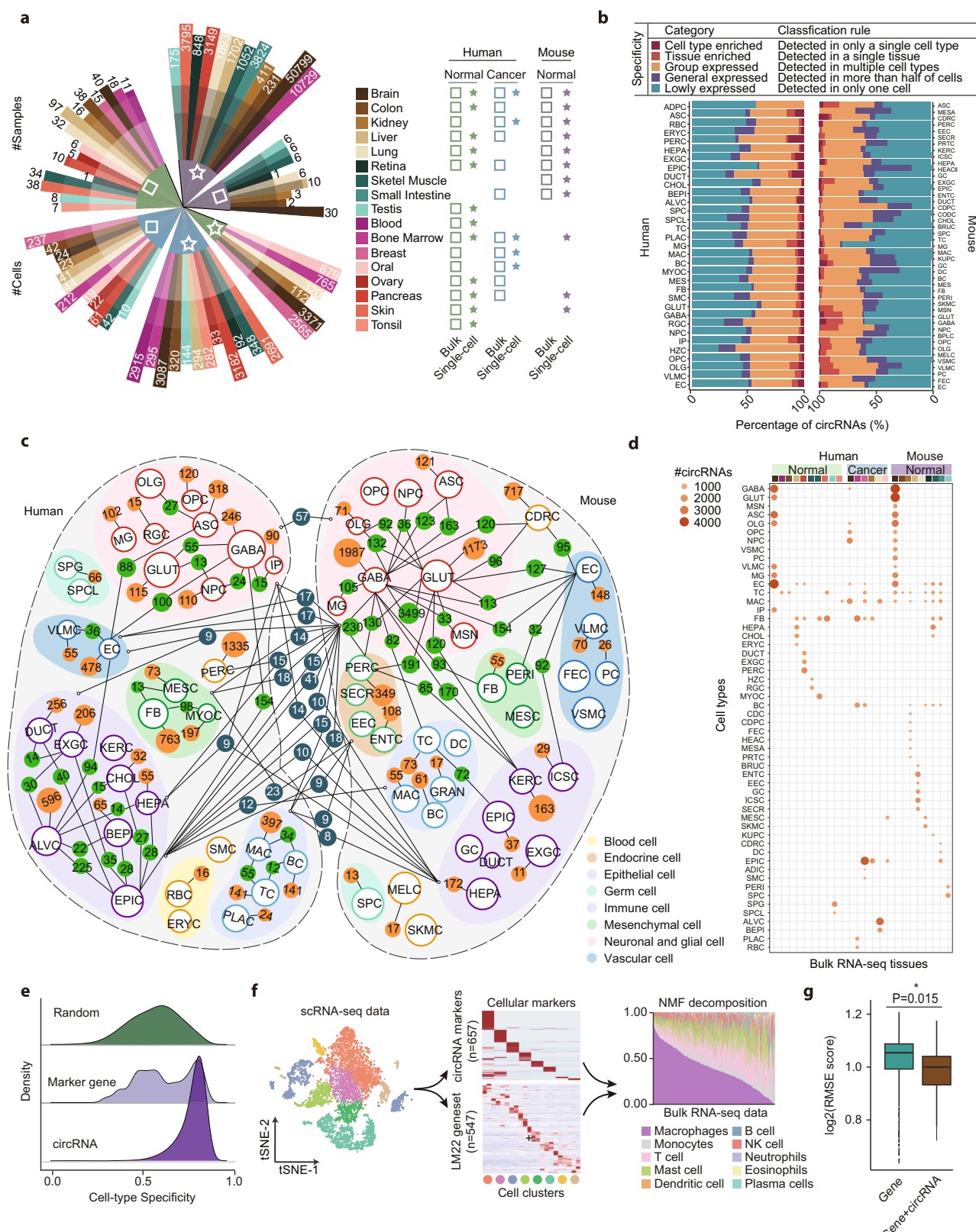

circRNAs during the human and mouse embryo development process. Aside from the maternal effect of circRNAs reported in previous studies[10,41], we further demonstrated the dramatic increase of circRNA expression during late MZT stages, which indicated the strong activation of zygotic circRNAs in pre-implantation embryos.

The circRNA expression in tumor samples has been extensively elucidated using bulk RNA-seq datasets. However, the results are often affected by the cancer-to-normal cell ratios among the studied tumor specimens, where the difference of tumor purity between samples could result in biased or false-positive results. In contrast to the well-known role of ciRS-7 as an oncogene[69], a

**Fig. 5 Exploring cell-type-specific circRNAs as biomarkers for cell composition deconvolution. a** Statistics of collected bulk RNA-seq and scRNA-seq datasets. **b** The number of cell-type enriched, tissue enriched, group enriched, general enriched, and lowly expressed circRNAs in each tissue. **c** The network plot showing the overlap of circRNAs expressed in each cell type. Only cell type enriched, tissue enriched, and group expressed circRNAs were included. Cell type nodes were clustered by their function and origin and annotated with different background colors. **d** The number of circRNAs detected in different cell types and bulk RNA-seq datasets. The x and y axis represents bulk RNA-seq and cell type in scRNA-seq data, respectively, and the size of points indicates the number of circRNAs simultaneously detected in both bulk and single-cell RNA-seq data. **e** The cell-type specificity of circRNAs, marker genes, and 1000 randomly selected genes. **f** Decomposing tumor-infiltrating immune cells using CIBERSORT with the LM22 gene set and marker circRNAs that are highly expressed (≥3 fold higher than all other cell types) in each cell type. Both circRNA- and gene-based deconvolution results were integrated to the 10 immune cell types identified in the scRNA-seq cohort. **g** Performance of cell type decomposition using marker circRNAs and genes ($n = 879$ independent bulk RNA-seq samples), respectively. The y axis represents the root-mean-squared error (RMSE) by CIBERSORT, which indicates the accuracy of the deconvolution result. All center lines in the box plots indicate the median values, and box limits indicate the upper and lower quartiles of plotted values. The upper and lower whiskers indicate the largest and smallest values within the range of 1.5x IQR from the box limits. *$P < 0.05$, Wilcoxon rank-sum test (two-sided). Source data are provided as a Source Data file.

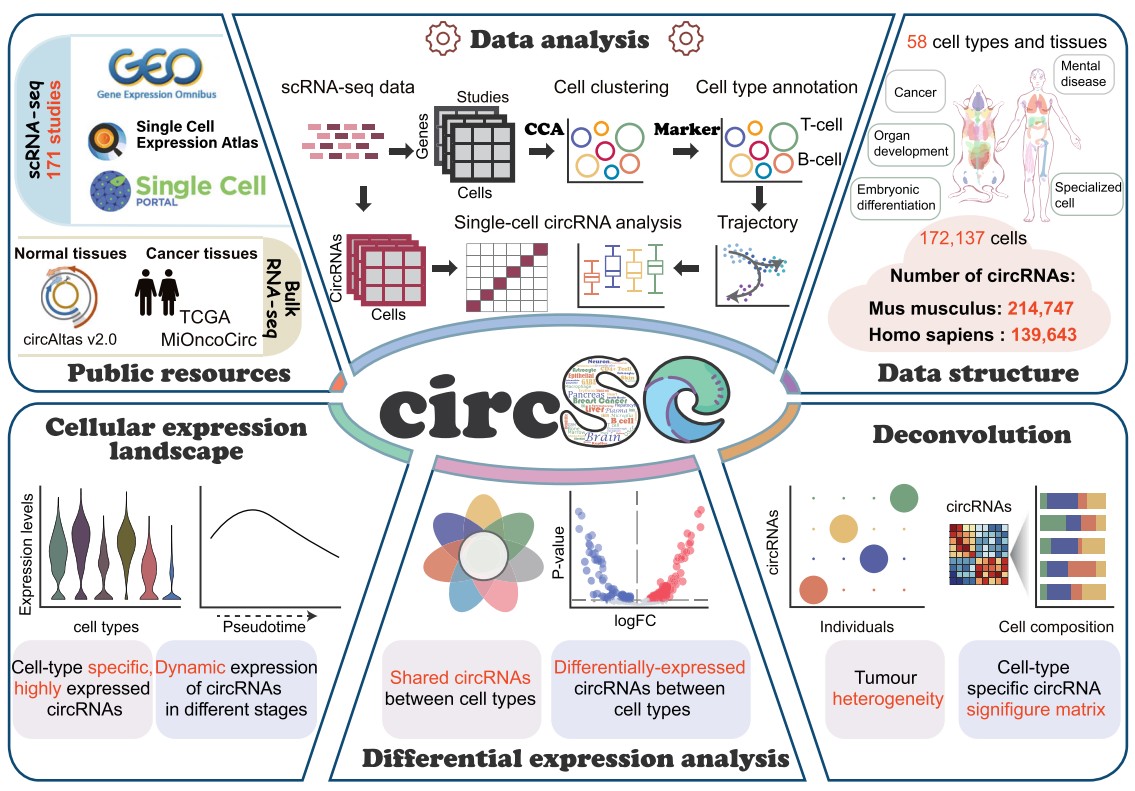

**Fig. 6 The construction and functionalities of the online portal circSC.** circSC consists of the cellular expression profiles of 354,390 circRNAs detected from 172,137 single cells, covering 58 cell types and tissues. circSC provides an intuitive interface for users to browse, search, and visualize the expression of circRNAs in various studies and cell types. It also integrates differential expression analysis between different cell types and the intra- and inter-heterogeneity of circRNAs between tumor patients. The circRNA signature matrix of tumor-infiltrating immune cells can be directly accessed, and users can input circRNA expression profiles from bulk RNA-seq data for deconvolution analysis. The anatogram is plotted using gganatogram[86].

recent study has experimentally validated that the expression of the ciRS-7 is absent in stromal tumor cells but highly expressed in stromal cells within tumors[12]. Thus, the investigation of circRNAs at the single-cell level has become an emerging aspect in studying circRNA function in tumor genesis and metastasis. In this study, we comprehensively investigated the expression landscape circRNAs in 20 breast carcinoma patients and demonstrated the heterogeneity of circRNAs between lesions and cell types. We utilized the EMT-score to measure the differentiation state of cells during the EMT process and further revealed the distinct changing pattern of normal and carcinoma cells. Finally, we also investigated the intertumoral heterogeneity of circRNAs between patients with different lineage stages. The circRNAs exhibited a similar expression pattern from primary and metastasis tumors from the same patient but have disparate expression patterns between patients. These high heterogeneities

of circRNAs suggested the importance of single-cell level investigation of circRNAs, which provides an important basis to understand the role of circRNAs during tumorigenesis.

Our previous studies have revealed the highly specific expression of circRNAs in different tissues and species[7,9,70,71]. Here, we explored the highly cell-specific expression of circRNAs at single-cell resolution and identified 12,625 circRNAs that were only detected in one cell type. Moreover, we generated the circRNA reference of 8 immune cell types and validated that the cell composition deconvolution results using circRNAs as cell-type signatures have better accuracy compared to that using gene markers only, which suggested the emerging role of these circRNAs as promising biomarkers in profiling tumor-infiltrating immune cells. This study further explores the cellular landscape and reveals high cell specificity of circRNAs in human and mouse samples, which largely expands our understanding of circRNA

biogenesis during complex biological processes. Therefore, we developed the circSC database to investigate the circRNA in single-cell resolution, which will provide a useful platform for the circRNA community. Nevertheless, the construction of the full panorama of circRNAs is still limited by the low circRNA capture efficiency in state-of-art scRNA-seq methods, and the performance of cell-type decomposition can also be affected by the relatively low expression of these cell-type specific circRNAs in bulk RNA-seq samples. At the same time, recent nanopore sequencing based strategies like isoCirc[72] and CIRI-long[73,74] have been proved to be able to capture lowly expressed circRNAs in bulk RNA-seq libraries with high efficiency. However, further comparison demonstrates these methods still have inadequate capacity in detecting cell-type specific circRNAs (Supplementary Fig. 6). Taken together, our study has demonstrated the highly specific expression of circRNAs at an unprecedent resolution, which suggests the emerging importance of developing further single-cell or spatial level sequencing technologies specifically for detecting circRNAs.

## Methods

**Single-cell RNA-seq dataset collection.** Full-length single-cell RNA sequencing datasets were collected from publicly available resources across multiple tissues and cell types of human and mouse samples. Raw sequencing data were downloaded from the Single Cell Expression Atlas (https://ebi.ac.uk/gxa/sc/home) and the Gene Expression Omnibus (https://ncbi.nlm.nih.gov/geo) using the SRA-Toolkit (v2.9.4). Metadata information of these datasets was retrieved from the corresponding literature. To ensure the effective capture of circRNAs, only full-length and high-resolution single-cell transcriptome sequencing methods including MATQ-seq[16], Quartz-seq[17], RamDA-seq[18], SMARTer[19], Smart-seq[20], Smart-seq2[21], SUPeR-seq[10], and Tang's method[22] were included. The detailed information of study accession number and cell numbers of the collected cohort was provided in the Supplementary Data 1.

**Single-cell RNA-seq analysis and integration.** For analysis of scRNA-seq data, the human reference genome (GRCh38) and mouse reference genome (GRCm38) were downloaded from the GENCODE project. Then, raw sequencing reads were aligned using HISAT2 (v2.0.5)[23], and StringTie (v1.2.4)[24] was performed for gene quantification. Next, a quality control step was implemented by Scater (v1.18.6)[25] to filter high confidence cells, where the appropriate thresholds of library size, gene expression values, mitochondrial reads, and the total amount of mRNA indicators in each study were estimated by *perCellQCMetrics* function. Afterward, the outlier cells were identified based on median-absolute-deviation (MAD) using *isOutlier* function.

Then, we used Seurat (v4.0.2)[26] to perform downstream analysis including normalization, batch effect removal, dimensional reduction, clustering, and data visualization. The anchor-based canonical correlation analysis (CCA) method in the Seurat package was performed for dataset integration and batch effect correction. Then, the integrated data was adopted to highly variable genes analysis, principal component analysis (PCA), neighborhood graph, and cell type clustering using the default parameters. Considering the inconsistency between different datasets, the normalized expression of mRNA and circRNAs was calculated by the size factor from integrated data performed by Scater[25].

**Cell type annotation.** Cell clusters were annotated based on canonical cell markers from published literature (Supplementary Table 3) and databases including CellMarker[63] and PanglaoDB[64]. Then, annotation results are curated using the SingleR (v1.4.1)[65] algorithm with various reference datasets (Blueprint/ENCODE, human primary cell atlas, Novershtern hematopoietic data, Monaco immune data, and Database of Immune Cell Expression). The curated annotation results are determined by combining both results from our pipeline and the original studies. The CopyKAT (v1.0.4)[52] workflow is used to determine normal cells and carcinoma cells with aneuploid rearrangement. The list of abbreviations for cell type names is listed in Supplementary Table 4.

**CircRNA detection and quantification.** For circRNA analysis, we used bwa (v0.7.12)[75] for split-mapping of raw reads, then the CIRI2 (v2.0.6)[27] and CIRIquant (v1.1)[29] pipeline was performed for circRNA identification and quantification. Then, stringent circRNAs were further filtered with a threshold of 2 supporting BSJ reads in the whole single cell dataset. The circRNA expression levels are measured using counts per million mapped reads (CPM). To eliminate the batch effect between different datasets, the number of supporting reads of each circRNA is normalized using size factor from gene normalization results, and the expression matrix at single-cell level is generated as output. Then, the circRNA expression profile in various tissues are aggregated by summing the expression

value of circRNA in each cell. Finally, *FindAllMarkers* function in Seurat was used for differential expression analysis.

**Rerverse transcription PCR (RT-PCR) validation.** To validate the reliability of circRNA detection in the scRNA-seq data. Outward primers were specifically designed to validate the back-spliced junction sequence of 12 randomly selected cell-type specific circRNAs that were detected in less than 10 cells. For RT-PCR, total RNA from the brain of one healthy adult mice (C57BL/6, female, 17 weeks) was isolated using TRIzol (Invitrogen, 15596026 and 15596018), and the quality was assessed with Qsep 100 Bio-Fragment Analyzer (BiOptic). The linear RNAs were digested with 20 U of RNase R (Lucigen, RNR07250) in a 50 μl reaction for 30 min according to a previous study[76], and ribosomal RNA was removed using KAPA RiboErase Kit (Human/Mouse/Rat, KK8481) according to the manufacturer's instructions. Here, a 2.2x RNA Clean XP (Beckman, A63987) cleanup was performed after each step. Finally, cleaned RNA was reverse transcribed using random primers and the Hifair® II 1st Strand cDNA Synthesis Kit (Yeasen, 11121ES60) following the manufacturer's instruction. Then RT-PCR experiments of 12 circRNAs were performed using Rapid Taq Master Mix (Vazyme P222) under the following conditions: 95 °C for 3 min; 35 cycles of 95 °C for 15 s, 55 °C for 15 s, and 72 °C for 60 s; 72 °C for 10 min. Finally, the sequences of PCR products were determined using Sanger sequencing. All sequences of primers and PCR products were supplied in the Supplementray Table 2.

**Trajectory analysis.** For branching trajectory and pseudo-time analysis, Monocle 2 (v.2.8.2)[77] was performed on scRNA-seq data to reveal the cell differentiation state. Cluster information was extracted from the Seurat results, and high variable genes were selected to determine the transition state or development process.

**Public circRNA databases and bulk RNA-seq data.** To validate the circRNAs detected in scRNA-seq data, a total of 10 public circRNA resources, including circAtlas (v2.0)[7], circbank[78], circBase[5], CIRCpedia (v2)[6], CircRic[15], circRNADb[79], MiOncoCirc (v2.0)[62], deepbase (v2.0)[80], TCSD[8] and CSCD[81] were collected. The circRNA coordinate was converted to the hg38/mm10 genome using liftover, and all circRNAs were integrated for downstream analysis. The length of the full-length assembled circRNAs in circAtlas was extracted for comparison. The bulk RNA-seq data of normal and tumor samples were downloaded from circAtlas and MiOncoCirc database and analyzed using the same method described above.

**Gene ontology enrichment analysis.** Gene set enrichment analysis against Gene Ontology pathways was performed by the ClusterProfiler (v4.0)[82] and Enrichr[83] software. The significant GO terms were filtered by a threshold of $p < 0.05$ values using the modified Fisher's exact test.

**Maternal and zygotic circRNAs cluster.** Hierarchical clustering in embryo development for gene and circRNA was performed based on fuzzy c-means clustering by Mfuzz (v2.50.0)[42].

**Cell specificity calculation.** The cell specificity of gene and circRNAs was calculated using the following equation:

$$T = \frac{\sum_{i=1}^{n}(1 - \hat{x}_i)}{n-1}; \hat{x}_i = \frac{x_i}{\max_{0 \le x \le 1}(x_i)} \quad (1)$$

Where $x_i$ is the average expression value of genes or circRNAs in different cells, and $n$ is the number of tissues or cell types.

**Cell composition inference.** All carcinoma cells were integrated and clustered as described above, then the epithelial cells, fibroblast, and endothelial cells were removed from the cluster results. The remaining cells were clustered again, and cell clusters were annotated to different immune cell types (macrophages, monocytes, T cells, mast cells, dendritic cells, B-cell, NK cells, neutrophils, eosinophils, and plasma cells). The circRNA signature was then filtered using the following criteria: (1) circRNAs expressed in at least 2 cell types; (2) circRNAs exhibited a significantly higher expression in one cell type than the others. The LM22 gene signature matrix was downloaded from the CIBERSORT webserver[68], and cell composition deconvolution results were aggregated to the cell types described above. The RMSE and correlation from CIBERSORT results were used for comparison.

**Statistics & reproducibility.** No statistical method was used to predetermine sample size. No data were excluded from the analyses and all analyses were not randomized.

**Reporting summary**. Further information on research design is available in the Nature Research Reporting Summary linked to this article.

## Data availability

The cellular expression results of circRNAs reported in this study have been deposited in the Genome Sequence Archive[84], China National Center for Bioinformation under accession number "PRJCA009653". The RNA-seq datasets used for circRNA identification are listed in the Supplementary Data 1. Source data have been deposited in "zenodo [https://doi.org/10.5281/zenodo.6528434]". All other relevant data supporting the key findings of this study are available within the article and its Supplementary Information files or from the corresponding author upon reasonable request. Source data are provided with this paper.

## Code availability

The analysis pipeline is available at the "circSC" module in "circAtlas [http://circatlas.biols.ac.cn]" and in the "Github repository [https://github.com/bioinfo-biols/Code_for_circSC]"[85].

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

## Acknowledgements
This work was supported by grants to F.Z. from the National Natural Science Foundation of China [32130020, 32025009, 91940306] and National Key R&D Project [2021YFA1300500] and to J.Z. from the National Key R&D Project [2021YFA1302000].

## Author contributions
F.Z. conceived the project. W.W. and J.Z. analyzed the data. X.C. and Z.C. performed the experiments. W.W. designed the database. J.Z., W.W., and F.Z. wrote the manuscript. The authors read and approved the final manuscript.

## Competing interests
The authors declare no competing interests.
