## [Peer Review File · Nature Communications]

REVIEWER COMMENTS

Reviewer #1 (Remarks to the Author):

Wu et al. present a comprehensive in silico analysis of circular RNAs (circRNAs) in published single cell studies. The authors select a subset of 171 single cell studies in human and mouse that employs library designs that are able to capture circRNAs. Given this vast amount of data the team presents more than 200k novel circRNAs expressed in different cell types. However, the pure amount of generated data combined with the fine-grained resolution makes it hard to verify these finding in vitro or in vivo. The circSC portal provides expressed circRNAs for a few cells of a one cell type in one experiment – but is the minuscule amount of circRNAs of functional relevance? Is it even possible to perform qRT-PCR for verification? How do I manipulate a specific circRNA from the database in a biological system?

1. For the web portal, a domain name should be used instead of an IP address. The danger of infrastructure changes making the service unreachable is lower if a (sub)domain is used than can be redirect to a new service when the server changes.
2. The authors state that 254,590 cells are used for analyses, but later it becomes clear that is pre-filtering, yielding only ~172k usable cells. Since this is the actual number of cells, I would remove the 254k earlier as it sounds like all of those are used.
3. Moreover, after filtering that makes ~1,000 cells per experiment, which is not that much compared to other single cells datasets. Maybe I missed it, but the authors should include this information somewhere, as larger datasets can easily yield tens of thousands of cells alone.
4. I am not sure how much sense Fig 1A make for the human subsets where cell types don't fit in the graphics anymore, a table might be a more helpful presentation of the data.
5. Is there a reason why for example Fig 1D is a box plot and 1E a violin plot? For 1E the mean value can only be seen when extremely zoomed in. Specifically, for 1E it would be interesting to only look at the ~80% full length circRNAs from circAtlas compared to the single cell circRNAs. Since the single cell circRNAs are not fully sequenced (or maybe I missed that?) this might be an overestimation of the circRNA length.
6. Fig 2C shows that the majority of circRNAs shown (65%) is not part of any database and a such counts as novel. However, Figure 2H impressively shows that the mean number is hardly over, so I'd like to know what the mean value of cells is? The authors write 90% are in less than 10 cells. However, ~71% are in 1 cell (Fig 1J) which, given that on average an experiment has 1,000 cells seems low.
7. Given the expression of these ultra-rare circRNAs, I'd like to know if the authors made any attempt to verify some of these. My point here being: what is the biological relevance of these extremely lowly

abundant circRNAs? As the authors state, bulk RNA-seq would not pick them up – but what technique would, single cell aside?

8. The authors outline the methods for data processing and the seurat package in parts in the manuscript and the Reporting Summary. However, due to the massive amount of data produced and the complete focus in silico work, the data processing methods should be more clearly outlined, optimally as code available on GitHub or GitLab. Otherwise, important preprocessing steps are not reproducible.

Reviewer #2 (Remarks to the Author):

In this study, Wu et al collected 171 full-length single-cell RNA-seq datasets to explore the cellular landscape of circRNAs in human and mouse tissues. Using this large-scale data, they characterized the cell-type-specific expression pattern of circRNAs brain samples, developing embryos, and breast tumors, which are all very important biological/biomedical fields with great research interest for a broad research community. Strikingly, they showed that uniquely expressed circRNAs could perform well in tumor infiltrating immune cell composition deconvolution. Finally, the authors developed an online portal for exploring circRNAs at an unprecedented resolution. Overall, the study is comprehensive and insightful, the data resource is useful for the broad research community, and it deserves to be published as soon as possible. The reviewer has a few comments/suggestions:

1. Line 179 – the authors may consider using Spearman correlation to calculate the correlation between circRNAs and circRNA hosting genes or RBPs instead of Pearson correlation.
2. Line 338- To use circRNAs as biomarker for cell-type classification, will it be possible that some circRNAs could not be detected simply due to low expression, so that affect the cell-type classification? The authors should further discuss this limitation.
3. Line 373 – what is RMSE?
4. Some references are out-of-dated. For example, reference 69 (CSCD2) /70 (ClusterProfiler 4.0) has updated version. The authors may check carefully for the most updated references.

Reviewer #3 (Remarks to the Author):

Although circular RNAs (circRNAs) are non-coding, recent evidence suggests that they are widely expressed in mammalian cells with cell type and/or tissue-specific expression patterns, and they have been shown to have important functional roles that might have a large impact on cellular phenotypes and disease. In spite of their potential importance, there is limited understanding of circRNAs' functional roles in a wide variety of settings.

The authors conduct a comprehensive and careful integrative analysis of over 150 full-length single-cell RNA-seq datasets to identify, and then validate, hundreds of thousands of circRNAs in human and mouse. They also demonstrate utility of this type of data by identifying cell-specific expression patterns (e.g. brain circRNAs in inhibitory vs. excitatory neurons; maternal and zygotic circRNAs in distinct stages of embryonic development). An online portal (circSC) is also provided in an effort to make results widely available.

Major

The authors previously developed CircAtlas (a database containing over a million circRNAs from 1070 vertebrate transcriptomes). CircAtlas was published in *Genome Biology* and judging from citations (>100 since 2020) is gaining some traction. The results in CircAtlas were derived from bulk RNA-seq data. Here, the authors focus on characterizing circRNAs in human and mouse tissue at the single cell level. They then demonstrate cell-type-specific expression pattern of circRNAs in brain samples, developing embryos, and breast tumor tissue. An online portal (circSC) is provided to facilitate analysis of circRNAs at the single cell level.

The biological results here demonstrate utility of circRNAs in a few settings. However, the main contribution of this work is in making the circRNAs and associated results widely available. Consequently, the database described here is particularly important. Unfortunately, it is difficult to use. For example, I can see a list of circRNAs expressed in a specific tissue, but there is no option to download, and no results (e.g. DE genes) associated with them. I expect these results are available, but it's not clear where. Furthermore, there is no manual detailing how to conduct the most common queries. The manual link simply points the user to an email address. These types of challenges will dramatically limit impact.

It seems like the results presented here could be added to CircAtlas, especially given that users are already familiar

with that portal. I am not sure what advantage is gained by not merging the two.

How do the results presented here compare with those in isoCirc where the authors generate a comprehensive catalog of 107,147 full-length circRNA isoforms across 12 human tissues and one human cell line (HEK293)(Xin et al., isoCirc catalogs full-length circular RNA isoforms in human transcriptomes, *Nature Communications*, 2021)?

This work should be discussed in detail with advantages of the submitted work highlighted. As isoCirc is not considering

single cell data, I expect there will be important differences. However, it is not clear how extensive the differences will be particularly in tissues/samples that are homogeneous (i.e. where you don't gain much by single-cell over bulk).

Minor

In methods, the authors note that "The detailed information of study accession number and cell numbers of the collected cohort are provided in data. S1". This reference is confusing. I had assumed it was referring to Table S1, which does not have this info. The first xlsx sheet in the supplementary data, which does have this info, is not labelled S1.

Furthermore, the total number of cells from the xlsx sheet is 254,490 and not 254,590 as stated in the text. I'm

assuming that is typo.

The authors note that Fan et al. is limited because they focused on "studying the maternal effect of circRNAs in 69 mouse embryo cells..." While I agree that Fan et al. is limited in focus relative to the study submitted here, I am not sure where the 69 is coming from. Fan et al. note that they considered "69 mouse mature oocyte and preimplantation embryo samples" where each sample contains at least thousands of cells. Please clarify.

REVIEWER COMMENTS

Reviewer #1 (Remarks to the Author):

Wu et al. present a comprehensive in silico analysis of circular RNAs (circRNAs) in published single cell studies. The authors select a subset of 171 single cell studies in human and mouse that employs library designs that are able to capture circRNAs. Given this vast amount of data the team presents more than 200k novel circRNAs expressed in different cell types. However, the pure amount of generated data combined with the fine-grained resolution makes it hard to verify these findings in vitro or in vivo. The circSC portal provides expressed circRNAs for a few cells of a one cell type in one experiment – but is the minuscule amount of circRNAs of functional relevance? Is it even possible to perform qRT-PCR for verification? How do I manipulate a specific circRNA from the database in a biological system?

Response: Thanks for these insightful comments. We agree with the reviewer that the biological relevance of these rare cell-type-specific circRNAs and even most circRNAs detected in bulk RNA-seq samples are difficult to verify. In the revised manuscript, we have performed RT-PCR and Sanger sequencing to validate the BSJ sequences of 12 randomly selected cell-type specific circRNAs in RNase R treated RNAs from mouse brain samples, which demonstrate the high reliability of circRNA detection results. Besides, we have also provided the cell-type specific circRNAs in the updated version of the circSC portal and added additional information including their expression profiles in circAtlas, miRNA/RBP binding sites and co-expression networks, etc. Thus, users can easily obtain the expression landscape of circRNAs they interested in and explore the possible biological functions using both circSC and circAtlas databases. All the comments raised by the reviewer have been seriously considered. Please refer to the following responses for details.

1. For the web portal, a domain name should be used instead of an IP address. The danger of infrastructure changes making the service unreachable is lower if a (sub)domain is used than can be redirect to a new service when the server changes.

Response: Thanks for this kind suggestion. We have integrated this portal to circAtlas as an individual module (<http://circatlas.biols.ac.cn>) to protect the real IP address and update the URL in the revised manuscript accordingly.

2. The authors state that 254,590 cells are used for analyses, but later it becomes clear that is pre-filtering, yielding only ~172k usable cells. Since this is the actual number of cells, I would remove the 254k earlier as it sounds like all of those are used.

Response: As suggested, we have updated the description in line 63 of the revised manuscript.

3. Moreover, after filtering that makes ~1,000 cells per experiment, which is not that much compared to other single cells datasets. Maybe I missed it, but the authors should include this information somewhere, as larger datasets can easily yield tens of thousands of cells alone.

Response: As suggested, we have added the description in lines 98-100, and the exact number of filtered cells of each experiment has been included in the revised Supplementary Data. It should be noted that only full-length scRNA-seq studies were included here, which often have much fewer cell numbers than 10x scRNA-seq. Although 3'-end scRNA-seq methods like Drop-seq and 10x Genomics' Chromium can capture thousands of cells in a single experiment, they are not suitable for studying circRNAs that lack 3' tails.

4. I am not sure how much sense Fig 1A make for the human subsets where cell types don't fit in the graphics anymore, a table might be a more helpful presentation of the data.

Response: Thanks for the suggestion. Considering that a detailed table may not be easily fitted in the panel of Fig. 1a, we have included the number of cell types in each dataset in the revised Supplementary Data.

5. Is there a reason why for example Fig 1D is a box plot and 1E a violin plot? For 1E the mean value can only be seen when extremely zoomed in. Specifically, for 1E it would be interesting to only look at the ~80% full length circRNAs from circAtlas compared to the single cell circRNAs. Since the single cell circRNAs are not fully sequenced (or maybe I missed that?) this might be an overestimation of the circRNA length.

Response: Thanks for the kind suggestion. We must explain that the original Fig. 1e is already plotted using the full-length circRNAs, and we have updated the revised Fig. 1e to the box plot as suggested. Here, all circRNAs from the circAtlas database were divided into two subsets, including 730,657 circAtlas-specific circRNAs and 112,075 circRNAs that were shared between circAtlas and the scRNA-seq cohorts. Then, the length of the fully reconstructed circRNA sequences from circAtlas (619,060 "circAtlas" and 103,758 "overlap" circRNAs) was extracted from the circAtlas database for further comparison. We have included the detailed description in the figure legend and Methods section to avoid further confusion.

Figure. 1 (e) The length of fully assembled sequence of 619,060 circAtlas-specific and 103,758 circRNAs shared between circAtlas and the scRNA-seq cohort.

6. Fig 2C shows that the majority of circRNAs shown (65%) is not part of any database and a such counts as novel. However, Figure 2H impressively shows that the mean number is hardly over, so I'd like to know what the mean value of cells is? The authors write 90% are in less than 10 cells. However, ~71% are in 1 cell (Fig 1J) which, given that on average an experiment has 1,000 cells seems low.

Response: Thanks for the comment. We assume that the reviewer is referring to Fig. 1c and Fig. 1h which demonstrate the number and expressed cells of overlapping circRNAs between single-cell datasets and other bulk RNA-seq based databases. Specifically, “novel” circRNAs are expressed in an average of 1.48 cells (median = 1 cell), while the overlapping “known” circRNAs are generally expressed in 7.54 cells (median = 2 cells). Besides, both novel (mean = 7.85 BSJ) and known circRNAs (mean = 7.26 BSJ) are similarly supported, which also suggests the high confidence of these novel circRNAs. Thus, the high cell-level specificity expression of these novel circRNAs should be the major reason for their absence in the bulk RNA-seq datasets.

Besides, as explained in the #3 comment above, we have tried our best to collect as many full-length scRNA-seq datasets as possible. We believe with the further development of single-cell RNA-seq techniques, we will be able to expand the scale of our study in the future.

7. Given the expression of these ultra-rare circRNAs, I'd like to know if the authors made any attempt to verify some of these. My point here being: what is the biological relevance of these extremely lowly abundant circRNAs? As the authors state, bulk RNA-seq would not pick them up – but what technique would, single cell aside?

Response: As suggested, we first performed RT-PCR of 12 cell-type specific circRNAs (expressed in less than 10 cells) in rRNA depleted, RNase R treated mouse brain RNA samples. Then, Sanger sequencing was performed to validate back-spliced junction sequences of these circRNAs. Please refer to the revised Supplementary Table. 2 for detailed information.

Meanwhile, we believe that it's hard to determine the biological relevance even for most circRNAs from the bulk RNA-seq datasets, not to mention these rare circRNAs detected at single-cell resolution. However, our manuscript provides the important landscape of circRNAs at the single-cell level, which may serve as an important resource for the whole circRNA community. Moreover, we have included the number of supporting bulk RNA-seq datasets of 12,625 cell-type specific circRNAs reported in the revised Supplementary Fig.5, which demonstrate the feasibility of using these circRNAs as cell-type markers in bulk RNA-seq datasets.

Supplementary Fig. 5 Expression of cell-type specific circRNAs in the circAtlas database. (a) The number of cell-type specific circRNAs that are reported in bulk RNA-seq based databases. **(b)** The number of expressed tissues and samples of 6,139 cell-type specific circRNAs in the circAtlas database. The z-axis represents the number of expressed cells in the scRNA-seq cohort.

Finally, we also included the results of two nanopore-based circRNA sequencing strategies, isoCirc and CIRI-long, in the revised Supplementary Fig. 6. Although these methods have improved sensitivity in circRNA detection, they are still unable to effectively characterize these circRNAs detected in scRNA-seq libraries. Thus, we believe that most current bulk-based sequencing protocols are deficient in detecting these cell-specific or cell-type-specific circRNAs.

Supplementary Fig. 6 Comparison of scRNA-seq methods against isoCirc and CIRI-long. (a) Overlap of circRNAs detected in human samples from isoCirc, circAtlas and the scRNA-seq datasets. **(b)** Overlap of circRNAs detected in each tissue. **(c)** Boxplots indicate the number of expressed cells of each subset. **(d)** Boxplots indicate the number of isoCirc BSJ reads of each subset. **(e)** Overlap of circRNAs detected in mouse brain samples from CIRI-long, circAtlas and the scRNA-seq datasets. **(f)** Boxplots indicate the number of CIRI-long BSJ reads of each subset. **(g)** Boxplots indicate the number of expressed cells of each subset.

8. The authors outline the methods for data processing and the `seurat` package in parts in the manuscript and the Reporting Summary. However, due to the massive amount of data produced and the complete focus in *in silico* work, the data processing methods should be more clearly outlined, optimally as code available on GitHub or GitLab. Otherwise, important preprocessing steps are not reproducible.

Response: Thanks for the suggestion. We have uploaded the customized pipeline and detailed instructions in the “Download” section in circSC to prompt the reproducibility of our work.

Reviewer #2 (Remarks to the Author):

In this study, Wu et al collected 171 full-length single-cell RNA-seq datasets to explore the cellular landscape of circRNAs in human and mouse tissues. Using this large-scale data, they characterized the cell-type-specific expression pattern of circRNAs brain samples, developing embryos, and breast tumors, which are all very important biological/biomedical fields with great research interest for a broad research community. Strikingly, they showed that uniquely expressed circRNAs could perform well in tumor infiltrating immune cell composition deconvolution. Finally, the authors developed an online portal for exploring circRNAs at an unprecedented resolution. Overall, the study is comprehensive and insightful, the data resource is useful for the broad research community, and it deserves to be published as soon as possible. The reviewer has a few comments/suggestions:

Response: We greatly appreciate the reviewer's comments on the novelty and significance of our study.

1. Line 179 – the authors may consider using Spearman correlation to calculate the correlation between circRNAs and circRNA hosting genes or RBPs instead of Pearson correlation.

Response: Thanks for the suggestion. We have updated Fig. 2 g & h using Spearman correlation and updated the description in the main text accordingly.

2. Line 338- To use circRNAs as biomarker for cell-type classification, will it be possible that some circRNAs could not be detected simply due to low expression, so that affect the cell-type classification? The authors should further discuss this limitation.

Response: Thanks for the suggestion. We agree with the reviewer that some cell-type specific circRNAs could not be detected due to the low level of expression. As shown in Supplementary Fig. 5a, 6,623 (52.5%) of the 12,625 cell-type specific circRNAs were also reported in the bulk RNA-seq databases. Notably, these circRNAs were mutually detected in a variety of bulk RNA-seq libraries of different tissue and samples (Supplementary Fig. 5b), which proved the feasibility of using these circRNAs as cell-type markers in bulk RNA-seq datasets. We have also discussed this limitation in lines 343-348 and lines 463-474 of the revised manuscript.

Supplementary Fig. 5 Expression of cell-type specific circRNAs in the circAtlas database. (a) The number of cell-type specific circRNAs that are reported in bulk RNA-seq based databases. **(b)** The number of expressed tissues and samples of 6,139 cell-type specific circRNAs in the circAtlas database. The z-axis represents the number of expressed cells in the scRNA-seq cohort.

3. Line 373 – what is RMSE?

Response: The performance of deconvolution results was assessed by log-scale root-mean-square error (RMSE) in the CIBERSORT output. We have updated the description of RMSE in the main text to avoid further confusion.

4. Some references are out-of-dated. For example, reference 69 (CSCD2)/70 (ClusterProfiler 4.0) has updated version. The authors may check carefully for the most updated references.

Response: Thanks for the suggestion, we have carefully revised the manuscript and updated all the references.

Reviewer #3 (Remarks to the Author):

Although circular RNAs (circRNAs) are non-coding, recent evidence suggests that they are widely expressed in mammalian cells with cell type and/or tissue-specific expression patterns, and they have been shown to have important functional roles that might have a large impact on cellular phenotypes and disease. In spite of their potential importance, there is limited understanding of circRNAs' functional roles in a wide variety of settings.

The authors conduct a comprehensive and careful integrative analysis of over 150 full-length single-cell RNA-seq datasets to identify, and then validate, hundreds of thousands of circRNAs in human and mouse. They also demonstrate utility of this type of data by identifying cell-specific expression patterns (e.g. brain circRNAs in inhibitory vs. excitatory neurons; maternal and zygotic circRNAs in distinct stages of embryonic development). An online portal (circSC) is also provided in an effort to make results widely available.

Response: We greatly appreciate the reviewer's comments on the novelty and significance of our study.

Major

The authors previously developed CircAtlas (a database containing over a million circRNAs from 1070 vertebrate transcriptomes). CircAtlas was published in Genome Biology and judging from citations (>100 since 2020) is gaining some traction. The results in CircAtlas were derived from bulk RNA-seq data. Here, the authors focus on characterizing circRNAs in human and mouse tissue at the single cell level. They then demonstrate cell-type-specific expression pattern of circRNAs in brain samples, developing embryos, and breast tumor tissue. An online portal (circSC) is provided

to facilitate analysis of circRNAs at the single cell level.

The biological results here demonstrate utility of circRNAs in a few settings. However, the main contribution of this work is in making the circRNAs and associated results widely available. Consequently, the database described here is particularly important. Unfortunately, it is difficult to use. For example, I can see a list of circRNAs expressed in a specific tissue, but there is no option to download, and no results (e.g. DE genes) associated with them. I expect these results are available, but it's not clear where. Furthermore, there is no manual detailing how to conduct the most common queries. The manual link simply points the user to an email address. These types of challenges will dramatically limit impact.

Response: Thanks for the suggestion. We have improved the usability of the circSC portal in many aspects and included detailed instructions on interpreting and downloading analysis results. In the updated version of circSC, users can browse and download the list of tissue-specific and cell-specific circRNAs, as well as obtain the link of the single-cell expression pattern of circRNAs in the circAtlas database. Moreover, we also provided links to the single-cell expression pattern of circRNAs in the circAtlas database, which enables convenient queries of the single-cell expression landscape for the circAtlas users.

It seems like the results presented here could be added to CircAtlas, especially given that users are already familiar with that portal. I am not sure what advantage is gained by not merging the two.

Response: Thanks for the suggestion. The circAtlas database is composed of samples from normal tissues only, while the circSC portal has included both tumor and normal scRNA-seq datasets. Thus, we think that it might be not appropriate to simply merge these two databases. Instead, we have migrated the circSC portal to a submodule under circAtlas (<http://circatlas.biols.ac.cn>), so users can easily access the single-cell expression data from the circAtlas database. Besides, we have updated both circSC and circAtlas to include the corresponding links of bulk- and single-cell expression patterns of the overlapping circRNAs.

How do the results presented here compare with those in isoCirc where the authors generate a comprehensive catalog of 107,147 full-length circRNA isoforms across 12 human tissues and one human cell line (HEK293) (Xin et al., isoCirc catalogs full-length circular RNA isoforms in human transcriptomes, *Nature Communications*, 2021)? This work should be discussed in detail with advantages of the submitted work highlighted. As isoCirc is not considering single cell data, I expect there will be important differences. However, it is not clear how extensive the differences will be particularly in tissues/samples that are homogeneous (i.e. where you don't gain much by single-cell over bulk).

Response: Thanks for the comment. As suggested, we have included the comparison of the circRNAs identified in the single-cell cohort and two recently published full-length circRNA resources, isoCirc (PMID: 33436621) and CIRI-long (PMID: 33707777) in Supplementary Fig. 6.

However, we must explain that it's hard to compare circRNAs in homogeneous tissue between the isoCirc and scRNA-seq datasets. As mentioned by Xin et al. "Each human tissue RNA was a

pooled sample from tissues of multiple donors”, which makes the heterogeneity between samples from different individuals evitable. Besides, Zhong et al. (PMID: 29089539) have also revealed the high heterogeneity of circRNA expression, even in the same group of HEK293T cells. Thus, we focused on the overall circRNA detection preference in the isoCirc and CIRI-long datasets for further comparison.

Supplementary Fig. 6 Comparison of scRNA-seq methods against isoCirc and CIRI-long. (a) Overlap of circRNAs detected in human samples from isoCirc, circAtlas and the scRNA-seq datasets. (b) Overlap of circRNAs detected in each tissue. (c) Boxplots indicate the number of expressed cells of each subset. (d) Boxplots indicate the number of isoCirc BSJ reads of each subset. (e) Overlap of circRNAs detected in mouse brain samples from CIRI-long, circAtlas and the scRNA-seq datasets. (f) Boxplots indicate the number of CIRI-long BSJ reads of each subset. (g) Boxplots indicate the number of expressed cells of each subset.

For the comparison with isoCirc, the overlap between circRNAs detected in the human samples from isoCirc, circAtlas and the scRNA-seq cohort was plotted. As shown in Supplementary Fig. 6a, a considerable fraction of circRNAs was mutually covered by all three methods. Besides, 39.3% (35,248 of 89,637) and 54.0% (75,436 of 139,643) of circRNAs were uniquely detected in the isoCirc and scRNA-seq datasets, respectively. The number of overlapped circRNAs in different tissues was also demonstrated in Supplementary Fig. 6b. We reason that although isoCirc utilized RNase R treatment and circular ligation steps to enrich circRNAs in the bulk tissue samples, its ability in detecting rare cell-specific circRNAs is still limited. Thus, circRNAs were further divided into disjoint subsets based on the Venn diagram, and the number of expressed cells in the scRNA-seq cohort and supporting BSJ reads in isoCirc data were assessed in the Supplementary Fig. 6c and 6d. As expected, scRNA-seq specific subsets 1&2 were expressed in a significantly lower number of cells compared to subsets 3&4, and the isoCirc specific subsets 5&6 were also supported

by fewer BSJ reads. Subsequently, we further compared the circRNAs detected in the mouse brain samples using CIRC-long, circAtlas, and the scRNA-seq data, where a similar pattern was also observed (Supplementary Fig. 6 e-g).

Taken together, these results demonstrated that while isoCirc and CIRC-long utilized different experimental treatments to increase sensitivity in detecting lowly expressed circRNAs, they are still not able to capture these ultra-rare cell-type specific circRNAs effectively. Meanwhile, the single-cell sequencing methods have demonstrated their unique superiority in detecting cell-specific circRNAs. However, we agree that the detection of circRNAs at single-cell resolution is still limited by the relatively low circRNA capture efficiency of present scRNA-seq techniques and have added the discussion of this limitation as suggested by other reviewers.

Minor

In methods, the authors note that "The detailed information of study accession number and cell numbers of the collected cohort are provided in data. S1". This reference is confusing. I had assumed it was referring to Table S1, which does not have this info. The first xlsx sheet in the supplementary data, which does have this info, is not labelled S1.

Response: Thanks for the suggestion. We have updated the reference in the main text.

Furthermore, the total number of cells from the xlsx sheet is 254,490 and not 254,590 as stated in the text. I'm assuming that is typo.

Response: Thanks for pointing this out. One Smart-seq2 study was mistakenly counted in the original table, and the correct number is 204,253 in the revised Supplementary Data. Besides, we have also replaced the numbers in the main text with 172,137 high-confidence cells as suggested by Reviewer #1.

The authors note that Fan et al. is limited because they focused on "studying the maternal effect of circRNAs in 69 mouse embryo cells..." While I agree that Fan et al. is limited in focus relative to the study submitted here, I am not sure where the 69 is coming from. Fan et al. note that they considered "69 mouse mature oocyte and preimplantation embryo samples" where each sample contains at least thousands of cells. Please clarify.

Response: Thanks for the suggestion. We have corrected the description in lines 49-51 accordingly.

REVIEWERS' COMMENTS

Reviewer #1 (Remarks to the Author):

Thank you for addressing my questions. I have one follow up comment regarding the RT-PCR, see below:

Author's Response:

As suggested, we first performed RT-PCR of 12 cell-type specific circRNAs (expressed in less than 10 cells) in rRNA depleted, RNase R treated mouse brain RNA samples. Then, Sanger sequencing was performed to validate back-spliced junction sequences of these circRNAs. Please refer to the revised Supplementary Table. 2 for detailed information.

1) The selection of brain tissue might have direct influence on the probability of picking up of circRNAs due to the generally higher expression values of circRNAs in brain tissue.

2) It would be helpful if the authors would provide details regarding the RT-PCR such as Ct values, as this would also be indicative of the level of available transcript.

Reviewer #3 (Remarks to the Author):

The authors addressed my comments, and I have no further comment.

Reviewer #4 (Remarks to the Author):

The authors have addressed all of my comments. Congratulations on this nice work

Reviewer #1 (Remarks to the Author):

Thank you for addressing my questions. I have one follow up comment regarding the RT-PCR, see below:

1) The selection of brain tissue might have direct influence on the probability of picking up of circRNAs due to the generally higher expression values of circRNAs in brain tissue.

Response: Thanks for the comments. We agree with the reviewer that circRNAs are more likely to be captured in brain tissues, and that's the reason for choosing brain samples for PCR validation. For other tissues with lower circRNA expression levels, the probability of non-specific amplification could increase due to the extremely low copy number of templates. Here, we only want to focus on validating the existence of these cell-type specific circRNAs captured by single-cell based methods, so the brain tissue with higher expression circRNA levels were selected for experimental validation.

2) It would be helpful if the authors would provide details regarding the RT-PCR such as Ct values, as this would also be indicative of the level of available transcript.

Response: Thanks for the suggestion. Here, non-specific bands were observed for 9 of 12 cell-type specific circRNAs, which prevented us from performing further qRT-PCR experiments to examine the expression level of these circRNAs. Besides, considering the high cell-type heterogeneity in bulk tissues, the Ct values in bulk samples might not be suitable for direct comparison against scRNA-seq results. Thus, only Sanger sequencing was performed to validate the sequence of each amplified band, and the successfully validated BSJ sequences were shown in Supplementary Table 2.

Reviewer #3 (Remarks to the Author):

The authors addressed my comments, and I have no further comment.

We greatly appreciate the reviewer's comments on the novelty and significance of our study.

Reviewer #4 (Remarks to the Author):

The authors have addressed all of my comments. Congratulations on this nice work!

Thanks for the reviewer's comment on helping us improve the manuscript.